# Alkyl Chain Engineering of Low Bandgap Non-Fullerene Acceptors for High-Performance Organic Solar Cells: Branched vs. Linear Alkyl Side Chains

**DOI:** 10.3390/polym14183812

**Published:** 2022-09-12

**Authors:** Youngwan Lee, Telugu Bhim Raju, Hyerim Yeom, Peddaboodi Gopikrishna, Kwangmin Kim, Hye Won Cho, Jung Woo Moon, Jeong Ho Cho, Jin Young Kim, BongSoo Kim

**Affiliations:** 1Department of Chemistry, Ulsan National Institute of Science and Technology (UNIST), 50 UNIST-gil, Ulsan 44919, Korea; 2School of Energy and Chemical Engineering, Ulsan National Institute of Science and Technology (UNIST), 50 UNIST-gil, Ulsan 44919, Korea; 3Department of Chemical and Biomolecular Engineering, Yonsei University, Seoul 03722, Korea; 4Graduate School of Carbon Neutrality, Ulsan National Institute of Science and Technology (UNIST), 50 UNIST-gil, Ulsan 44919, Korea; 5Graduate School of Semiconductor Materials and Device Engineering, Ulsan National Institute of Science and Technology (UNIST), 50 UNIST-gil, Ulsan 44919, Korea

**Keywords:** organic solar cells, non-fullerene acceptor, power conversion efficiency, alkyl chain engineering

## Abstract

In this work, we report the synthesis and photovoltaic properties of IEBICO-4F, IEHICO-4F, IOICO-4F, and IDICO-4F non-fullerene acceptors (NFAs) bearing different types of alkyl chains (2-ehtylhexyl (EH), 2-ethylbutyl (EB), *n*-octyl (O), and *n*-decyl (D), respectively). These NFAs are based on the central indacenodithiophene (IDT) donor core and the same terminal group of 2-(5,6-difluoro-3-oxo-2,3-dihydro-1*H*-inden-1-ylidene)malononitrile (IC-2F), albeit with different side chains appended to the thiophene bridge unit. Although the side chains induced negligible differences between the NFAs in terms of optical band gaps and molecular energy levels, they did lead to changes in their melting points and crystallinity. The NFAs with branched alkyl chains exhibited weaker intermolecular interactions and crystallinity than those with linear alkyl chains. Organic solar cells (OSCs) were fabricated by blending these NFAs with the p-type polymer PTB7-Th. The NFAs with appended branched alkyl chains (IEHICO-4F and IEBICO-4F) possessed superior photovoltaic properties than those with appended linear alkyl chains (IOICO-4F and IDICO-4F). This result can be ascribed mainly to the thin-film morphology. Furthermore, the NFA-based blend films with appended branched alkyl chains exhibited the optimal degree of aggregation and miscibility, whereas the NFA-based blend films with appended linear alkyl chains exhibited higher levels of self-aggregation and lower miscibility between the NFA molecule and the PTB7-Th polymer. We demonstrate that changing the alkyl chain on the π-bridging unit in fused-ring-based NFAs is an effective strategy for improving their photovoltaic performance in bulk heterojunction-type OSCs.

## 1. Introduction

Organic solar cells (OSCs) have been considered one of the most promising alternative renewable energy sources since the past two decades owing to their advantages, such as high mechanical flexibility, light weight, low-cost, and facile solution processability [1,2]. It is imperative to develop new solution-processable organic semiconducting materials to realize OSCs with high power conversion efficiencies (PCEs) [3,4]. In general, an organic photoactive layer contains an appropriate *p*-type donor polymer and an *n*-type small-molecule acceptor [5,6,7]. Over the past few decades, fullerene derivatives were commonly used as *n*-type acceptors in OSCs because of their high electron affinity and isotropically good electron-transportation ability. However, they have several drawbacks including high cost, weak light absorption in the ultraviolet (UV)-visible region, limited synthetic tunability, and morphological instability [8]. Non-fullerene acceptors (NFAs) have been developed to overcome these limitations, and they have emerged as mainstream materials for fabricating OSCs. They have several advantageous features, such as relatively low synthesis cost, facile structural modification, tunable energy levels, and intense, broad light absorption [9]. NFA-incorporated OSCs have recently exhibited impressive PCEs, exceeding 18% for binary and 19% for ternary devices [10,11,12,13].

High-performing NFAs generally possess an acceptor–donor–acceptor (A-D-A)-type structure. Their energy levels and light harvesting ability from the UV-visible to near-infrared (NIR) can be tuned by changing the electron-donating core (D) units, alkyl side chains, and flanked acceptor (A) groups. Considerable efforts have been dedicated to the development of planar ladder-type D units, such as indacenodithiophene (IDT) and indacenodithieno [3,2-*b*]thiophene (IDTT), to improve PCEs. The rigid extended fused-ring structure of NFAs prevents rotational disorder, reduces reorganization energy, and enhances charge carrier mobility. Moreover, the effective conjugation length and strong delocalization of π-electrons leads to a narrow optical bandgap, which maximizes effective utilization of NIR light and UV-visible light [5,14].

In addition to the molecular core structure, the type and position of alkyl side chains in NFAs can significantly influence their solubility, thin-film morphology, molecular energy level, charge mobility, and eventually their photovoltaic performance [15,16]. For instance, the introduction of too bulky side chains on the molecular backbone might destroy the ordered π–π stacking owing to steric hindrance. The effect of *p*-type polymers possessing various alkyl chains on the photovoltaic properties of NFAs has been investigated extensively [17]. However, studies concerning the effects of alkyl substituents on NFAs are relatively rare [18,19,20,21,22,23]. For instance, Li et al. compared ITIC with *para*-alkylphenyl (*p*-ITIC) with ITIC with *meta*-alkylphenyl (*m*-ITIC) and found that *m*-ITIC exhibited high crystallinity and electron mobility, which resulted in a PCE of up to 11.77% [18]. Marks et al. developed ITIC derivatives (ITIC-CX) by means of side-chain modification with *n*-propyl, *n*-hexyl, and *n*-nonyl. They demonstrated that the PCEs of the PBDB-T:ITIC-C9 and PBDB-TF:ITIC-C9 blended films were higher than those of the ITIC-C6 and ITIC-C3 films because of the enhanced structural order and superior charge transport [19]. Similarly, Bo et al. modified the ITIC molecule as ITIC-SC6, ITIC-SC8, and ITIC-SC2C6 by replacing 4-(alkoxy)phenyl with a 4-(alkylthio)phenyl side chain. The PCE of the PBDBST:ITIC-SC2C6-based OSCs (9.16%) was higher than that of the PBDBST:ITIC-SC6 based OSCs (7.27%) and PBDBST:ITIC-SC8 based OSCs (7.79%). Their results indicated that ITIC-SC2C6 possessed good solubility and facilitated the formation of favorable fibrous nanostructures in the blend system [20]. Recently, Hou et al. replaced 2-ethylhexyl with 2-butyloctyl in the Y6 molecule [21] to obtain BTP-4F-12, which has a more pronounced face-on orientation and a higher electron mobility than the Y6 molecule. The OSC based on the PBDB-TF:BTP-4F-12 blend film exhibited an impressive PCE of 16.4% [22]. Yan et al. modified the Y6 molecule with different alkyl chains on the nitrogen atoms of the pyrrole motifs to develop N-C11 (undecanyl substituted on the nitrogen), N3 (n- propyl on the nitrogen), and N4 (n-butyl on the nitrogen). They found that the N3 molecule with the branched alkyl chain at the 3rd position exhibited the best performance with a PCE of 16.74% because of its optimal solubility, as well as favorable electronic and morphological characteristics [23]. These results demonstrate that side-chain modification is a facile and efficient approach to tune the crystallinity, morphology, and photophysical properties of NFAs, with the aim of achieving high PCEs.

In this work, we design and synthesize a series of IRICO-4F NFAs bearing alkyl (R) side chains on the thiophene π-bridge, where R denotes 2-ethylhexyl (EH) in the case of IEHICO-4F, 2-ethylbutyl (EB) in the case of IEBICO-4F, *n*-octyl (O) in the case of IOICO-4F, and *n*-decyl (D) in the case of IDICO-4F (see the chemical structures in Figure 1). In the IRICO-4F molecule, 3-alkoxythiophene can lock ladder-type IDT donor aromatic units through intramolecular noncovalent interactions between the IDT thiophene S-atom and the alkoxy O-atom (S‧‧‧O) to form a planar structure. Bulky rigid phenyl substituents on the IDT backbone do not disrupt the planarity of the IDT core, in addition to preventing molecular aggregation. The 2-(5,6-difluoro-3-oxo-2,3-dihydro-1*H*-inden-1-ylidene)malononitrile (IC-2F) acceptor is selected as the terminal electron-withdrawing group with the aim of creating a good electron transport channel through the close π–π stacking. Moreover, the strong electronegativity of the fluorine atoms can enhance intramolecular charge transfer in the A-π-D-π-A-type acceptor and can reduce the optical bandgap, which is beneficial for enhancing the light-harvesting ability of the NFA molecule.

In thin-film absorption spectra, the IRICO-4F acceptors exhibit a narrow-bandgap of ~1.29 eV, with strong absorption in the 600–850 nm region, and a redshift of ~55 nm is observed compared to that of chloroform solution. As the length of the alkyl pendant increases, the solubility of the IRICO-4F gradually improves. To study the photovoltaic properties of IRICO-4F, inverted OSCs are fabricated by blending IRICO-4F with the PTB7-Th *p*-type polymer. Compared to the OSCs based on the NFAs with appended linear alkyl chains, those based on the NFAs with appended branched alkyl chains exhibited a higher PCE: 10.63% for PTB7-Th:IEBICO-4F and 10.64% for PTB7-Th:IEHICO-4F. The higher PCEs of the PTB7-Th:IEBICO-4F- and PTB7-Th:IEHICO-4F-based devices are attributed to the formation of optimal thin-film morphology and molecular packing. By contrast, the PTB7-Th:IOICO-4F films are characterized by high roughness with coarse phase separation, and the PTB7-Th:IDICO-4F films have pinholes, which decrease their PCEs: 9.93% for PTB7-Th:IOICO-4F and 8.98% for PTB7-Th:IDICO-4F. Our work demonstrates the importance of appending alkyl side chains for controlling solubility, intermolecular interaction, film morphology, and eventually, the photovoltaic property of the NFA molecules.

## 2. Results and Discussion

### 2.1. Synthesis of NFAs

Synthetic routes for preparing IEBICO-4F, IOICO-4F, and IDICO-4F are provided in Figure 1. 2-Bromo-3-alkoxythiophene 5-formaldehyde (5) was synthesized following the procedure reported in [24,25]. The key intermediates (7a, 7b, and 7c) were synthesized by Stille coupling of bis(stannyl) IDT and 2-bromo 5-formyl thienyl π-bridges with 3-alkoxy side chains (5a, 5b, and 5c). Knoevenagel condensation of the dialdehyde precursors (7a, 7b, and 7c) with IC-2F afforded the target NFAs, namely IOICO-4F, IDICO-4F, and IEBICO-4F in yields >75%. All of the newly synthesized compounds and intermediates were characterized with ^1^H-nuclear magnetic resonance (^1^H-NMR), ^13^C-NMR, and Matrix-Assisted Laser Desorption/Ionization Time-of-Flight (MALDI-TOF) analysis (Appendix A). These NFAs exhibited good solubility in chloroform, tetrahydrofuran, and chlorobenzene solvents (see Appendix A). Notably, commercial IEHICO-4F, too, was used in this study.

### 2.2. Optical and Electrochemical Properties

The ultraviolet/visible (UV/Vis) absorption properties of the NFAs were determined in dilute chloroform solution and thin films, as displayed in Figure 1b, and their optical data are summarized in Table 1. These four NFAs exhibited strong absorption bands in the range of 600–850 nm in chloroform solution, which can be ascribed to the intramolecular electron transfer transitions between the central IDT-core and the terminal IC-2F acceptor units. As shown in Figure 1b, in thin films, all of these NFAs exhibited redshifted absorption from 804 to 850–860 nm compared to their solution state, owing to strong π–π interactions in the thin films. The optical band gap of the NFAs was estimated to be ~1.28 eV in the film state. The extinction coefficients (ɛ_max_s) of the NFAs were measured in chloroform solution and thin films, and the related data are summarized in Table 1. All the NFAs show high absorption coefficients of 1.8 × 10^5^ M^−1^·cm^−1^ for IEBICO-4F, 1.6 × 10^5^ M^−1^·cm^−1^ for IEHICO-4F, 1.5 × 10^5^ M^−1^·cm^−1^ for IOICO-4F, and 1.8 × 10^5^ M^−1^cm^−1^ for IDICO-4F in solution state and 0.81 × 10^5^ M^−1^·cm^−1^ for IEBICO-4F, 0.93 × 10^5^ M^−1^·cm^−1^ for IEHICO-4F, 0.93 × 10^5^ M^−1^·cm^−1^ for IOICO-4F, and 0.93 × 10^5^ M^−1^·cm^−1^ for IDICO-4F in the film state.

The energy levels of the NFAs were determined by means of cyclic voltammetry (CV), and the curves are shown in Figure 1c. The highest occupied molecular orbital (HOMO) and lowest unoccupied molecular orbital (LUMO) energy levels were calculated as follows: E_HOMO/LUMO_ = −(eV*_onset,ox/red_* + 4.8 eV), where eV*_onset,ox/red_* is the onset potential of molecular oxidation or reduction with respect to the oxidation potential of the ferrocene molecule, and the corresponding data are listed in Table 1. The estimated HOMO/LUMO energy levels of IEBICO-4F, IEHICO-4F, IOICO-4F, and IDICO-4F were −5.37/−3.98, −5.41/−4.05, −5.40/−3.99, and −5.41/−3.99 eV, respectively. The electrochemical band gaps were 1.39, 1.36, 1.41, and 1.42 eV for IEBICO-4F, IEHICO-4F, IOICO-4F, and IDICO-4F, respectively. The energy level band diagram is shown in Figure 1d. These NFAs exhibited suitable energy levels and complementary absorption with the PTB7-Th donor polymer.

### 2.3. Thermal Properties

Differential scanning calorimetry (DSC) measurements were performed to investigate the phase transitions and crystallization properties of the NFAs. As shown in Figure 2, except for IEHICO-4F (*T*_m_ = 110.97 °C), the NFAs possessed higher melting points (*T*_m_) at 219.11 °C and 256.73 °C for IEBICO-4F, 277.32 °C for IOICO-4F, and 265.42 °C for IDICO-4F. Compared to the NFAs appended with branched alkyl chains, the NFAs appended with linear alkyl chains had higher *T*_m_ values, which clearly indicates that the intermolecular interaction of these NFAs is strong.

### 2.4. Photovoltaic Properties

To study the influence of the different alkyl chains on photovoltaic device performance, we fabricated a series of OSC devices by using IRICO-4F acceptors with an inverted structure of indium tin oxide (ITO)/ZnO/active layer (PTB7-Th:IRICO-4F)/V_2_O_5_/Ag. The PTB7-Th polymer was used as the donor material in this study because its energy levels and absorption properties are well matched with those of these NFAs. The active layers were prepared by spin coating the blend solution at room temperature. By varying the D:A weight ratios, the optimum blend ratio was identified when the D:A weight ratio was 1:1.2. The device engineering methods used herein are described in the Appendix A. The optimal current density–voltage (*J–V*) curves and external quantum efficiency (EQE) curves of these four OSCs are shown in Figure 3, and the corresponding photovoltaic parameters are summarized in Table 2. The optimized OSCs based on PTB7-Th:IEBICO-4F and PTB7-Th:IEHICO-4F exhibited the best PCEs of 10.63% and 10.64%, respectively. By contrast, the OSCs based on the NFAs appended with linear alkyl chains yielded lower PCEs of 9.93% for PTB7-Th:IOICO-4F and 8.98% for PTB7-Th:IDICO-4F because of the reduced short-circuit current density (*J*_SC_) and fill factor (FF) values. This difference can be explained based on morphological differences, which are described below [26,27]. The external quantum efficiency (EQE) spectra of the photovoltaic devices based on PTB7-Th:IRICO-4F are shown in Figure 3b. The IRICO-4F-based devices exhibited a broad photo response extending from 300 to 1000 nm, which closely corresponds to the absorption spectra of the blend films (Appendix A). The *J*_SC_ integral values of the IEBICO-4F-, IEHICO-4F-, IOICO-4F-, and IDICO-4F-based OSCs were estimated by integrating the EQE spectra, and the resulting integral values are 22.16, 21.29, 21.03, and 20.31 mA/cm^2^, respectively. The integrated *J*_SC_ values agree well with those obtained by performing *J–V* measurements.

To better understand the effect of the alkyl side chain on device performance and charge transport properties, electron and hole mobilities were measured using the space charge limited current (SCLC) method. Electron-only and hole-only devices were fabricated in the structures of ITO/ZnO/3-[6-(diphenylphosphinyl)-2-naphthalenyl]-1,10-Phenanthroline(Phen-NaDPO)/active layer/Phen-NaDPO/Ag and ITO/poly(3,4-ethylenedioxythiophene):poly(styrene sulfonic acid) (PEDOT:PSS)/active layer/MoO_3_/Al, respectively. *J–V* characteristics of hole-only and electron-only devices are shown in Appendix A; the extracted electron mobility (μ_e_) and hole mobility (μ_h_) values of the four blend films are summarized in Appendix A. For the calculation of the electron and hole mobilities, relative permittivity values of PTB7-Th:IRICO-4F films were obtained by electrochemical impedance spectroscopy (Appendix A). As shown in Appendix A, the calculated μ_h_/μ_e_ of the IEBICO-4F, IEHICO-4F, IOICO-4F, and IDICO-4F blend films are 1.11 × 10^−4^/1.01 × 10^−4^ cm^2^ V^−1^s^−1^ (ratio = 1.10), 1.23 × 10^−4^/1.30 × 10^−4^ cm^2^ V^−1^s^−1^ (0.95), 9.38 × 10^−5^/9.58 × 10^−5^ cm^2^ V^−1^s^−1^ (0.98), and 1.00 × 10^−4^/9.40 × 10^−5^ cm^2^ V^−1^s^−1^ (1.06), respectively. The higher carrier mobilities of the PTB7-Th:IEBICO-4F and PTB7-Th:IEHICO-4F-based devices are beneficial for efficient charge transport and charge extraction, which contributed to their higher *J*_SC_ and FF, compared to the PTB7-Th:IOICO-4F and PTB7-Th:IDICO-4F device.

### 2.5. Morphological Characterization

To determine the differences between the surface morphologies of the PTB7-Th:IRICO-4F blend films, atomic force microscopy (AFM) and transmission electron microscopy (TEM) measurements were performed, and the resulting images are displayed in Figure 4. The root-mean-square (RMS) roughness values of the PTB7-Th:IEBICO-4F, PTB7-Th:IEHICO-4F, PTB7-Th:IOICO-4F, and PTB7-Th:IDICO-4F blend films are 1.37, 1.42, 2.75, and 1.25 nm, respectively. The PTB7-Th:IEBICO-4F and PTB7-Th:IEHICO-4F blend films exhibited smooth and uniform surface with a small RMS roughness. Compared to the NFAs appended with linear alkyl chains, the NFAs appended with branched alkyl chains prevented film aggregation, leading to the generation of smooth and uniform films. In contrast, the PTB7-Th:IOICO-4F blend film had an uneven morphology with a high RMS roughness of 2.75 nm, which might be ascribed to the strong aggregation tendency of IOICO-4F. The PTB7-Th:IDICO-4F blend film also showed undesirable pinholes, despite a low RMS value of 1.25 nm, which is associated with its poor photovoltaic property. The results of a TEM analysis demonstrated that compared to the blend films based on the NFAs with appended linear alkyl chains, the blend films based on the NFAs with appended branched alkyl chains had a uniform, well-mixed morphology. Similar to the AFM results, the TEM analysis revealed the existence of larger domains with higher levels of aggregation for PTB7-Th:IOICO-4F. Moreover, the PTB7-Th:IDICO-4F blend film appeared to be highly miscible, with some notable pinholes. This formation of excessively large domains or pinholes was responsible for the lower FF values of these linear alkyl chain NFA-based devices.

The neat and blend films were subjected to two-dimensional grazing incidence wide-angle X-ray scattering (GIWAXS) measurements to investigate the molecular packing and orientation of IRICO-4F molecules. The GIWAXS images and the line-cut profiles are shown in Figure 5a,c respectively, for the IEBICO-4F, IEHICO-4F, IOICO-4F, and IDICO-4F neat films. The (100) scattering peak of the IEBICO-4F molecule is located at *q*_z_ = 0.35 Å^−1^ (*d* = 17.85 Å) in the out-of-plane (OOP) direction, which corresponds to a lamellar stacking peak. The (010) peak corresponding to the π–-π stacking is located at *q*_y_ = 1.78 Å^−1^ (*d* = 3.52 Å) in the in-plane (IP) direction. This feature indicates that the IEBICO-4F molecule assumes an edge-on orientation. Interestingly, the IEHICO-4F molecule exhibited a face-on orientation with improved crystalline features. The (100) peak was located at 0.33 Å^−1^ (*d* = 19.31 Å) in the IP direction, and the (010) peak was located at 1.82 Å^−1^ (*d* = 3.45 Å) in the OOP direction. The NFAs with linear alkyl chains exhibited an edge-on orientation. Notably, the IOICO-4F molecule had an edge-on orientation, even though it is a structural isomer of IEHICO-4F, thus demonstrating the strong influence of the type of alkyl side chains in the molecular packing. The (100) and (010) peaks of IOICO-4F were at 0.37 Å^−1^ (*d* = 16.97 Å) and 1.74 Å^−1^ (*d* = 3.60 Å), respectively. IDICO-4F had the strongest (100) peak at 0.38 Å^−1^ (*d* = 16.77 Å) among the edge-on oriented molecules, and it had weak high-order peaks as well; its weak (010) peak was located at 1.76 Å^−1^ (*d* = 3.56 Å). Considering all molecular packing characteristics of IRICO-4F, the lamella distances of the IEBICO-4F and IEHICO-4F neat films appended with branched alkyl chains were shorter than those of the IOICO-4F and IDICO-4F neat films appended with linear alkyl chains. This is because the branched alkyl chains possessed a shorter effective length owing to their high degree of conformational freedom. These results clearly indicate that the kind of alkyl chains which are appended on NFAs have a strong influence on their molecular packing. PTB7-Th film was also examined. The PTB7-Th polymer film exhibited a strong face-on orientation (Appendix A) with the lamellar stacking peak (100) and the π–π stacking (010) peak located at 0.26 and 1.62 Å^−1^, respectively. The main parameters of GIWAXS are summarized in Table 3.

When PTB7-Th was blended with IRICO-4F, the molecular orientation and packing behavior of the IRICO-4F NFAs changed significantly. As shown in Figure 5b,d, the strong π–π stacking (010) and lamellar stacking (100) peaks can be clearly observed in the OOP direction and IP direction, respectively, for the PTB7-Th:IRICO-4F blend films, suggesting that all the blend films showed preferential face-on orientation; PTB7-Th and IEHICO-4F both maintained their preferential face-on packing in the PTB7-Th:IEHICO-4F blend. It is of interest to note that in the blend films, the (100) peaks in the OOP direction of the other three NFAs disappeared as compared to their neat films, suggesting that the molecular orientation of these NFAs have changed from edge-on to face-on upon blending with PTB7-Th. In addition, PTB7-Th:IOICO-4F film presented significantly enhanced crystallinity with a narrow and stronger π–π stacking (010) peak, which is consistent with the higher surface roughness. In addition, after careful deconvolution of the (010) peaks in the blend films, the NFA/polymer peak area ratios of the PTB7-Th:IEBICO-4F and PTB7-Th:IEHICO-4F films were found to be 0.360 and 0.287, respectively, while those of the PTB7-Th:IOICO-4F and PTB7-Th:IDICO-4F films were found to be 0.595 and 0.547 (Table 4).

Together, these results and the morphological study results indicate that the IEBICO-4F and IEHICO-4F appended with branched alkyl chains tend to aggregate less around PTB7-Th polymer chains, whereas the IOICO-4F and IDICO-4F molecules appended with linear alkyl chains tend to aggregate more. In other words, the IEBICO-4F and IEHICO-4F molecules appended with branched alkyl chains are optimally miscible in PTB7-Th polymer, and for this reason, their photovoltaic properties are superior to those of the molecules appended with linear alkyl chains.

## 3. Conclusions

We synthesized a series of NFAs by appending branched or linear alkyl side chains on the 3-alkoxy thiophene bridge of IRICO-4F. These NFAs exhibited negligible differences in terms of optical properties and molecular energy levels. The NFAs appended with branched alkyl chains possessed lower melting points than those appended with linear alkyl chains. These results clearly indicate that the NFAs appended with linear alkyl chains have strong intermolecular interactions. Inverted OSCs based on PTB7-Th:IRICO-4F blends were fabricated. Among all devices, the devices based on the NFAs appended with branched alkyl chains exhibited PCEs >10.6%. By contrast, the OSCs based on the NFAs appended with linear alkyl chains exhibited lower PCEs. The results of morphological studies indicated that the PTB7-Th:IEBICO-4F and PTB7-Th:IEHICO-4F blend films had a smooth uniform surface with optimal miscibility, which improved *J*_SC_ and FF. By contrast, the PTB7-Th:IOICO-4F and PTB7-Th:IDICO-4F blend films displayed unfavorable aggregation with large domains or pinholes, respectively, resulting in lower *J*_SC_ and FF values. Our findings demonstrate that the modification of NFAs with suitable alkyl side chains is an important approach to further improve the PCE of OSCs.

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
