# Peer review of "Alkyl Chain Engineering of Low Bandgap Non-Fullerene Acceptors for High-Performance Organic Solar Cells: Branched vs. Linear Alkyl Side Chains"

_polymers, 2022, doi:10.3390/polym14183812_

Round 1

Reviewer 1 Report

Alkyl side chain engineering is an excellent tool for modulating the film morphology and photovoltaic properties of organic solar cells. In this work, the authors explored the effects of different kinds of alkyl side chains appended to non-fullerene acceptors (NFAs) on the morphology of bulk heterojunction (BHJ) films and performance of organic solar cells. I recommend its publication after some revisions by addressing the following issues.

1)       Page 4, in the section of “Optical and electrochemical properties”, line 154-155, the extinction coefficients of IEHICO-4F (93.34 ´ 105 cm−1) and IDICO-4F (89.34 × 105 cm−1) are one order of magnitude higher than commonly reported values in literatures (~105 cm−1), I suggest the authors check these values.

2)       The extinction coefficients reported on page 4 are different from those in Table 1.

3)       Page 4, line 156, there is no clear expression in reference 22 to support the statement that absorption coefficient is related to solubility.

4)       According to Figure 5a, IEHICO-4F shows the strongest crystallinity thus is supposed to show the highest melting point. But in the section of “Thermal properties” (page 5), the melting point of IEHICO-4F is the lowest. Some explanations should be added to rationalize this phenomenon.

5)       Page 7, in the section of “Morphological characterization”, the authors claimed “Compared to the NFAs appended with linear alkyl chains, the NFAs appended with branched alkyl chains prevented film aggregation, leading to the generation of smooth films”. But the blended film based on PTB7-Th and IDICO-4F with linear alkyl chains shows the smallest RMS roughness of 1.25 nm, which is contrary to the statement above.

6)       Page 2, to help the readers gain more knowledge about the development of OPVs and NFAs, some reviews are suggested to be cited (e.g., Acc. Chem. Res. 2021, 54, 132; Acc. Mater. Res. 2022, 3, 309; Adv. Energy Mater. 2022, DOI: 10.1002/aenm.202201087; Mater. Horiz. 2022, DOI: 10.1039/D2MH00376G).

Reviewer 2 Report

In this manuscript, the authors synthesized a series of IRICO-4F NFAs and fabricated inverted OSCs based on PTB7-Th:IRICO-4F blends. They found that devices using NFAs appended with branched alkyl chains exhibit higher PCEs than those with linear alkyl chains. By performing the morphological study on blend films, they showed that blend films based on branched alkyl chain NFAs display smooth and uniform surfaces with optimal miscibility, which contribute to higher JSC and FF in the devices. It is an interesting paper that can be published after revision. The following comments are needed to be addressed:

1.     For both hole-only and electron-only devices in SCLC measurements, the authors use only one hole transport layer (PEDOT:PSS) in hole-only devices and one electron transport layer (PEIE) in electron-only devices. In SCLC measurements, two HTLs and two ETLs are needed in hole-only and electron-only devices, respectively, to ensure ohmic contacts for one type of carriers and block the opposite type. Therefore, the mobility obtained from SCLC measurements in this work is not trustworthy. Please use two transport layers in the devices and perform SCLC measurements again.

2.     In Figure S18, the fitted curves and the data only agree in a small voltage range. In SCLC, J ~ V^2, please indicate the region where this relationship obeys. Why does this relation occur at different voltage ranges for different materials? Please specify the voltage ranges used for the SCLC fitting and the number of devices measured to get the standard deviation in Table S1.

3.     The authors mention IEBICO-4F, IOICO-4F, and IDICO-4F NFAs have good solubility in chloroform, tetrahydrofuran, and chlorobenzene solvents. Please provide the maximum solution concentration of each NFA (xx mg/ml) in each solvent.

4.     Please explain why εmax increase is higher in NFAs with branched alkyl chains than those with linear alkyl chains. That is, why is εmax of IEHICO-4F is ~ 2 times larger than that of IEBICO-4F while εmax values for IDICO-4F and IOICO-4F are about the same even though the carbon chain for IDICO-4F is longer than IOICO-4F?

5.     The authors specify the ɛr is equal to 3. How is this value obtained? Why is ɛr the same for different NFAs?

6.     Among four NFAs, the Tm of IEHICO-4F is ~ 111 °C, significantly lower than the Tm values of the rest NFAs. Please explain.

7.     The authors do not anneal active layers in the device. Annealing active layers is common to improve J-V performance in the devices. Please explain why thermal annealing is not necessary for these devices.

Reviewer 3 Report

Paper by Raju et al looks clear and propose new molecules to be used inside organic solar cell.

I don't have any specific comment to add, I recommend for publication after a careful language checking.

Author Response

We checked English one more time.

Round 2

Reviewer 2 Report

The authors addressed most of my concerns except for using ɛr equal to 3. Different organic materials have different ɛr values due to the polarizability of the molecule. The authors can easily determine ɛr from the capacitance value of the solar cell under reversed bias. The authors need to do this before the paper can be published.

Round 3

Reviewer 2 Report

The authors now use the “dielectric constant” obtained from ellipsometry to calculate the mobility. This is completely wrong. Ellipsometry is done at optical frequency and the J-V measurement is done near DC. Dielectric “constant” is a misnomer. Epsilon depends on the frequency. The value below 1 MHz and that at optical frequencies are drastically different. This is taught in Introduction to Materials Science classes and a quick web search will show the authors this fact. This revision reveals the authors’ ignorance about fundamental physical properties. This paper is still not acceptable for publication.
